# A Direct Current Measurement Method Based on Terbium Gallium Garnet Crystal and a Double Correlation Detection Algorithm

**DOI:** 10.3390/s19132997

**Published:** 2019-07-07

**Authors:** Yan Shen, Tong Chen, Wen-bin Yu, Jin-ming Ge, Yue Han, Fang-wei Duan

**Affiliations:** 1Department of Electrical Engineering, Harbin Institute of Technology, Harbin 150001, China; 2State Grid Liaoning Electric Power Research Institute, Shenyang 110006, China

**Keywords:** DC measurement, OCT, TGG crystal, signal processing, double correlation detection

## Abstract

When applying an optical current transformer (OCT) to direct current measurement, output signals exhibit a low signal-to-noise ratio and signal-to-noise band overlap. Sinusoidal wave modulation is used to solve this problem. A double correlation detection algorithm is used to extract the direct current (DC) signal, remove white noise and improve the signal-to-noise ratio. Our sensing unit uses a terbium gallium garnet crystal in order to increase the output signal-to-noise ratio and measurement sensitivity. Measurement errors of single correlation and double correlation detection algorithms are compared, and experimental results showed that this measurement method can control measurement error to about 0.3%, thus verifying its feasibility.

## 1. Introduction

DC transmission is an important means to realize interconnected power grids and transmission that has ultra-high voltage, large capacity, and low-loss for long distances. With the rapid development of high voltage direct current (HVDC) transmission systems, ultra-high voltage DC transmission projects are gaining popularity [1,2,3,4]. At present, the commonly used current transformer in DC power transmission systems is the electronic current transformer based on shunt. However, according to the principle of shunt, during the measurement, the line to be tested needs to be disconnected, power loss is large, and a measurement error will be generated due to heat generation [5,6,7]. In contrast, an OCT based on the Faraday magneto-optical effect can overcome these disadvantages. It does not need to disconnect the circuit under test. Simultaneously, it has the advantages of a large dynamic measurement range, wide frequency response range, and does not require an external power supply. Therefore, the application of OCTs in the measurement of a DC current has been rapidly developed [8,9,10,11].

However, many problems still persist in the DC current measurement of OCTs, such as a low signal-to-noise ratio (SNR) and the overlap of signal-to-noise bands. In order to solve these problems, Authors [12] used the spectrum migration measurement method. First, the DC signal to be measured is modulated into a high-frequency sinusoidal signal, and then low-frequency noise is filtered out through high-pass filtering in order to achieve signal-to-noise separation. After that, the signal to be measured is obtained using a signal demodulation method. Authors [13] studied and compared different demodulation methods based on the spectrum migration measurement method. It uses the synchronous demodulation method and the envelope detection method to demodulate the filtered signal and compares the results of the two. Authors [14] used a digital lock-in amplifier based on correlation detection principle to separate the signal and noise and detect the signals to achieve noiseless signal. These denoising methods can filter out noise in DC measurement to some extent. However, in signal processing, it is necessary to provide an additional demodulation signal or reference signal that is in accordance with the frequency of the modulation signal. Further, in hardware implementation, it is difficult to obtain modulation and demodulation signals with the same frequency. At the same time, these methods cannot effectively filter out white noise, and cause large errors in the measurement results.

In order to address the abovementioned problems, this paper proposes a DC optical measurement method based on a terbium gallium garnet (TGG) crystal and the double correlation detection algorithm. In this study, the basic principles and structure of the OCT are provided, the output characteristics of different sensor materials such as magneto-optical crystal and magneto-optical glass are compared, the signal-to-noise characteristics of the output signal is analyzed, and a double correlation detection algorithm is proposed for a weak signal which is submerged in noise. The measurement errors of the three denoising methods—synchronous demodulation, single correlation detection, and double correlation detection—are compared by simulation. The feasibility of this measurement method is verified by building a hardware experiment platform.

## 2. An OCT Based on TGG Crystal

The DC OCT uses the Faraday magneto-optical effect for current measurement. The schematic of the Faraday magneto-optic effect is shown in Figure 1. In a magneto-optical material, an applied magnetic field with a magnetic field intensity of *H* can deflect the polarization plane of the linearly polarized light, which propagates along the direction of the magnetic field. The deflection angle is as follows:(1)$$\theta =V\int_{l}\vec{H}d\vec{l}=V\cdot N\cdot i$$
where θ is the Faraday deflection angle of linearly polarized light, *V* is the Verdet constant of the magneto-optical material, *H* is the magnetic field intensity, *l* is the effective length vector of the linear polarized light when passing through materials under the magnetic effect, *N* is the number of rings of linearly polarized light around the current to be measured, and *i* is the current to be measured.

According to (1), the Faraday deflection angle is proportional to the current to be measured. Hence, the current to be measured can be obtained by measuring the deflection angle. However, the Faraday deflection angle of polarized light cannot be easily measured directly. Therefore, Marius’ law is usually used in OCT to convert the Faraday deflection angle into light intensity ease in measurement [15]. A natural light beam passes through a polarizer and becomes linearly polarized light with the light intensity $J_{0}$; according to Marius’s law, when it passes through an analyzer that forms an angle α with the polarization direction of the polarizer, the intensity of the emitted light is:(2)$$J_{1}=J_{0}\cos^{2}\alpha$$
where $J_{0}$ is the intensity of the linearly polarized light output after the polarizer, and $J_{1}$ is the polarized light output by the analyzer.

In order to obtain the maximum output intensity to increase the sensitivity of the OCT, α is usually 45°. Therefore, when the natural light output from the light source passes through the sensing unit, the light intensity output can be expressed as follows:(3)$$J_{1}=J_{0}\cos^{2}(45{}^{\circ}+\theta )=\frac{1}{2}J_{0}(1-\sin2\theta )\approx \frac{1}{2}J_{0}(1-2\theta )=\frac{1}{2}J_{0}(1-2NVi)$$

Further, after the light beam exits from the sensing unit and enters the photodetection unit, the light intensity signal can be converted into a voltage signal that can be easily measured. The voltage signal output by the photodetector is as follows:(4)$$V_{0}=U_{0}(1-2NVi)+n$$
where $V_{0}$ is the corresponding voltage signal of $J_{1}$, $U_{0}$ is the corresponding voltage signal of $\frac{1}{2}J_{0}$, and *n* is the noise produced by the photodetector.

The noises in DC OCT mainly arise from the photodetector, which are mainly white noise and low frequency noise with frequencies less than 1 kHz. Moreover, the currents to be measured are DC current and its harmonic components. This results in the overlap of the signal-to-noise bands, which makes it difficult to separate the signal and noise. At the same time, since the Faraday deflection angle is relatively small, the SNR of the DC OCT is very low. In engineering applications, the length of magneto-optical materials is shortened to reduce the influence of external temperature and linear birefringence on the OCT and improve its long-term operation stability [16]. From (1), it can be observed that reducing the length of the material will reduce the output SNR and even make the signal submerge in the noise. In order to solve the problem of a small Faraday deflection angle, this paper proposes a TGG crystal with a higher Verdet constant as the sensing unit. It can be noticed from (1) that increasing the Verdet constant V can increase the Faraday deflection angle and further improve the output SNR.

The TGG crystal is a high-performance magneto-optic crystal in the visible and near-infrared spectral range [17]. Compared with the magneto-optical glass commonly used in the OCT sensing unit, a magneto-optical crystal has the advantages of high Verdet constant, low absorption coefficient, and high laser damage threshold. In order to compare the sensing characteristics of different magneto-optical materials, Faraday rotation experiments are performed using magneto-optical crystals and anti-magnetic magneto-optical glass as sensing materials separately. The magneto-optical glass used in the experiment is the ZF-7 glass, which is commonly used in the current project and the magneto-optical crystal used is the TGG crystal. In this experiment, the sensing units are placed in a powered solenoid with a number of turns N = 600 respectively and the current is varied from 0–2 A. The result of the changes in the Faraday deflection angle obtained with the changing current is shown in Figure 2. It can be seen that using the TGG crystal can effectively increase the Faraday deflection angle as compared with the sensor unit using the ZF-7 magneto-optical glass. Therefore, the output SNR can be improved. At the same time, it can be seen from the slope of the fitting curve that using the TGG crystal can increase the measurement sensitivity of the sensing unit.

Although increasing the Faraday deflection angle can improve the output SNR, the effect of noise on the measurement results cannot be ignored. In order to address the problems of the overlapping of signal-to-noise bands and low SNR, using the randomness of noise in the OCT, this paper proposes a double correlation detection algorithm based on autocorrelation and cross-correlation operations in order to improve the measurement accuracy.

## 3. Double Correlation Detection Algorithm

Correlation detection technology is a kind of weak signal detection technology. It mainly uses the characteristics of signal correlation and random characteristics of noise to detect the signal. The correlation detection technology includes autocorrelation and cross-correlation algorithms [18,19,20,21]. Compared with the modulation and demodulation methods, the correlation detection algorithm does not require an additional demodulation signal. For periodic signals x(t) and y(t), the autocorrelation and cross-correlation functions are as follows:(5)$$R_{xx}(\tau )=E[x(t-\tau )x(t)]=\underset{T\to \infty }{\lim}\frac{1}{T}\int_{0}^{T}x(t-\tau )x(t)dt$$
(6)$$R_{xy}(\tau )=E[x(t-\tau )y(t)]=\underset{T\to \infty }{\lim}\frac{1}{T}\int_{0}^{T}x(t-\tau )y(t)dt$$
where *T* is the period of the periodic signal, τ is the delay.

Set the sine signal to be measured to s(t)=Asin(ωt+φ) and the noise signal to n(t); therefore, the input signal is:(7)x(t)=s(t)+n(t)=Asin(ωt+φ)+n(t)

Performing autocorrelation on (7):(8)$$\begin{array}{l} R_{xx1}(\tau )=E[x(t-\tau )x(t)]=E\{[s(t-\tau )+n(t-\tau )][s(t)+n(t)]\}\\ =E[s(t)s(t-\tau )]+E[n(t)n(t-\tau )]+E[s(t)n(t-\tau )]+E[n(t)s(t-\tau )]\\ =R_{ss}(\tau )+R_{nn}(\tau )+R_{sn}(\tau )+R_{ns}(\tau ) \end{array}$$
where $R_{ss}(\tau )$ is the autocorrelation function of the signal to be measured, $R_{nn}(\tau )$ is the autocorrelation function of the noise, $R_{sn}(\tau )$ and $R_{ns}(\tau )$ are the cross-correlation functions of noise and signal.

Because s(t) and n(t) are not related, we can consider $R_{sn}(\tau )=R_{ns}(\tau )=0$, and the autocorrelation function can be deduced as follows:(9)$$\begin{array}{l} R_{xx1}(\tau )=R_{ss}(\tau )+R_{nn}(\tau )\\ =\underset{T\to \infty }{\lim}\frac{1}{T}\int_{0}^{T}[s(t)s(t-\tau )]dt+R_{nn}(\tau )\\ =\underset{T\to \infty }{\lim}\frac{1}{T}\int_{0}^{T}[A\sin(\omega t+\varphi )A\sin(\omega (t-\tau )+\varphi )]dt+R_{nn}(\tau )\\ =\frac{A^{2}}{2}\cos(\omega \tau )+R_{nn}(\tau ) \end{array}$$

For broadband zero mean noise n(t), its autocorrelation function is mainly concentrated nearby τ=0. When τ→∞, it can be considered that $R_{nn}(\tau )=0$. Then, through $R_{xx1}(\tau )$, the amplitude of the signal to be measured can be extracted.

Phase-shifting the input signal x(t) by 90° can result in the following phase-shifted signal:(10)$$y(t)=A\cos(\omega t+\varphi )+n_{2}(t)=s_{2}(t)+n_{2}(t)$$
where $n_{2}(t)$ is the phase-shifted signal of n(t).

Performing cross-correlation on (7) and (10) gives:(11)$$\begin{array}{l} R_{xy1}(\tau )=E[x(t-\tau )y(t)]=E\{[s(t-\tau )+n(t-\tau )][s_{2}(t)+n_{2}(t)]\}\\ =E[s(t-\tau )s_{2}(t)]+E[n(t-\tau )n_{2}(t)]+E[s(t-\tau )n_{2}(t)]+E[n(t-\tau )s_{2}(t)]\\ =R_{ss_{2}}(\tau )+R_{nn_{2}}(\tau )+R_{sn_{2}}(\tau )+R_{ns_{2}}(\tau ) \end{array}$$
where $R_{ss_{2}}(\tau )$ is the cross-correlation function of s(t) and $s_{2}(t)$, $R_{nn_{2}}(\tau )$ is the cross-correlation function of the noise before which and after the phase shift, $R_{sn_{2}}(\tau )$ and $R_{ns_{2}}(\tau )$ are the cross-correlation functions of the noise and the signal before and after the phase shift.

Because the noise and signal are not related before and after the phase shift, we can consider $R_{sn_{2}}(\tau )=R_{ns_{2}}(\tau )=0$, and the cross-correlation function can be deduced as follows:(12)$$\begin{array}{l} R_{xy1}(\tau )=R_{ss_{2}}(\tau )+R_{nn_{2}}(\tau )\\ =\underset{T\to \infty }{\lim}\frac{1}{T}\int_{0}^{T}[s(t-\tau )s_{2}(t)]dt+R_{nn_{2}}(\tau )\\ =\underset{T\to \infty }{\lim}\frac{1}{T}\int_{0}^{T}[A\sin(\omega (t-\tau )+\varphi )A\cos(\omega t+\varphi )]dt+R_{nn_{2}}(\tau )\\ =-\frac{A^{2}}{2}\sin(\omega \tau )+R_{nn_{2}}(\tau ) \end{array}$$

For broadband zero mean noise n(t) and its phase-shifted signal $n_{2}(t)$, their cross-correlation function is mainly concentrated nearby τ=0, too. When τ→∞, it can be considered that $R_{nn_{2}}(\tau )=0$. Then, through $R_{xy1}(\tau )$, the amplitude of the signal to be measured can be extracted. Combining the autocorrelation operation and cross-correlation operation, the amplitude of the signal to be measured can be obtained as:(13)$$A=\sqrt{2\sqrt{\left(R_{xx1}^{2}+R_{xy1}^{2}\right)}}$$

Because only when τ is sufficiently large can we consider $R_{nn}(\tau )=R_{nn_{2}}(\tau )\approx 0$, in order to further reduce the impact of noise, we perform the double correlation detection operation on (9) and (12). It can be seen from (9) and (12) that for the sinusoidal signal to be measured and its phase-shifted signal, the autocorrelation function $R_{ss}(\tau )$ is a cosine function and the cross-correlation function $R_{ss_{2}}(\tau )$ is a sine function. Then, we can perform the autocorrelation and cross-correlation operations on $R_{xx1}(\tau )$ and $R_{xy1}(\tau )$ again, and the results are as follows:(14)$$\begin{array}{l} R_{xx2}(\tau )\approx \underset{T\to \infty }{\lim}\frac{1}{T}\int_{0}^{T}[\frac{A^{2}}{2}\cos(\omega t)\frac{A^{2}}{2}\cos(\omega (t-\tau ))]dt\\ =\frac{A^{4}}{8}\cos(\omega \tau ) \end{array}$$
(15)$$\begin{array}{l} R_{xy2}(\tau )\approx \underset{T\to \infty }{\lim}\frac{1}{T}\int_{0}^{T}-[\frac{A^{2}}{2}\cos(\omega (t-\tau ))\frac{A^{2}}{2}\sin(\omega t)]dt\\ =-\frac{A^{4}}{8}\sin(\omega \tau ) \end{array}$$
(16)$$A=\sqrt[{4}]{8\sqrt{R_{xx2}^{2}+R_{xy2}^{2}}}$$

Based upon the derivation results, further analysis is performed through Labview simulation. When the signal-to-noise ratio of the input signal is changed and the signal processing methods are the synchronous demodulation, single correlation detection algorithm and double correlation detection algorithm respectively, the measurement errors are compared. The measurement errors are listed in Table 1.

As shown in Table 1, when the SNR is fixed, double correlation detection can effectively filter out noise and reduce the measurement error as compared with the commonly used synchronous demodulation and single correlation detection method. At the same time, as the SNR increases, the measurement error decreases. In this measurement method, a double correlation detection algorithm is used for signal processing, and on the basis of this, the TGG crystal is used as a sensing material to increase the output SNR, which can, as a result, effectively reduce the measurement error and improve the DC measurement accuracy and sensitivity.

## 4. DC Measurement System Setup

The block diagram of the DC measurement system is shown in Figure 3. First, the optical source is modulated, and the modulating signal is:(17)$$u(t)=M_{1}+M_{2}\sin(\omega t)$$
where $M_{1}$ is the amount of DC modulation; $M_{2}$ is the amount of AC modulation; ω=2πf, where *f* is the modulating frequency. In order to ensure that the light-emitting diode (LED) emits light, it requires that $M_{1}>M_{2}$. After modulation, the input light intensity of the sensing unit $J_{i}$ becomes:(18)$$J_{i}=J_{0}(M_{1}+M_{2}\sin(\omega t))$$

In order to eliminate the effect of light intensity fluctuations on the measurement results, the signal output from the sensing unit is processed using the double light path method. The two analyzers are at 45° and −45° from the direction of the polarizer, and their outputs $J_{o1}$ and $J_{o2}$ are as follows:(19)$$J_{o1}=J_{i}\cos^{2}(45{}^{\circ}+\theta )\approx \frac{1}{2}J_{0}(M_{1}+M_{2}\sin(\omega t))(1-2\theta )$$
(20)$$J_{o2}=J_{i}\cos^{2}(-45{}^{\circ}+\theta )\approx \frac{1}{2}J_{0}(M_{1}+M_{2}\sin(\omega t))(1+2\theta )$$

After the output light passes through the photodetector, the light intensity signal is converted into a voltage signal as follows:(21)$$V_{o1}=U_{o}(M_{1}+M_{2}\sin(\omega t))(1-2\theta )+n_{1}(t)$$
(22)$$V_{o2}=U_{o}(M_{1}+M_{2}\sin(\omega t))(1+2\theta )+n_{2}(t)$$
where $V_{o1}$ and $V_{o2}$ are the voltage signals corresponding to $J_{o1}$ and $J_{o2}$, respectively, $U_{o}$ is the voltage signal corresponding to $\frac{1}{2}J_{0}$, $n_{1}(t)$ and $n_{2}(t)$ are the noises generated by the photodetectors.

The two-way signal passes through the band-pass filter whose center frequency is the modulating frequency, and the low-frequency noise and the DC component are filtered out to obtain:(23)$$V_{o1}^{\prime }=U_{o}M_{2}(1-2\theta )\sin(\omega t)+n_{1}^{\prime }(t)$$
(24)$$V_{o2}^{\prime }=U_{o}M_{2}(1+2\theta )\sin(\omega t)+n_{2}^{\prime }(t)$$
where $V_{o1}^{\prime }$ and $V_{o2}^{\prime }$ are the output voltages of filters, $n_{1}^{\prime }(t)$ and $n_{2}^{\prime }(t)$ are the remaining white noises after filtering.

$V_{o1}^{\prime }$ and $V_{o2}^{\prime }$ are phase-shifted and then the double-correlation detection operation is performed as described above, and their amplitudes are extracted, which can be noted as $V_{o1}^{\prime \prime }$ and $V_{o2}^{\prime \prime }$ respectively. Then, the deflection angle corresponding to the current to be measured can be obtained as follows:(25)$$\theta =\frac{V_{o2}^{\prime \prime }-V_{o1}^{\prime \prime }}{2(V_{o2}^{\prime \prime }+V_{o1}^{\prime \prime })}$$

It can be further obtained that:(26)$$i=\frac{\theta }{NV}=\frac{1}{2NV}\cdot \frac{V_{o2}^{\prime \prime }-V_{o1}^{\prime \prime }}{\left(V_{o2}^{\prime \prime }+V_{o1}^{\prime \prime }\right)}$$

Then the current can be measured.

## 5. Experiment and Result Analysis

The DC measurement experiment platform built in this study is shown in Figure 4.

Considering the filter effect of the filter circuit, the denoising effect, and the influence of the modulation frequency on the amplitude of the output signal, the modulation frequency is selected as 5 kHz. The modulated optical source includes an LED driving circuit and an LED. The LED driving circuit includes a sine-voltage-generating circuit, a voltage bias circuit, an adding circuit, and a voltage-controlled current source circuit. The sinusoidal generation circuit selects ICL8038 as the signal generation chip, which generates a sinusoidal voltage signal with a frequency of 5 kHz and an amplitude of 2 V. The bias voltage is superimposed onto the sine signal with an adding circuit. Then, the voltage-controlled current source circuit converts the voltage signal into a current signal with an amplitude of 20–60 mA. The sinusoidal current signal acts as a driving current for the LEDs. The LED adopts HFBR-1415. When the driving current is 20–60 mA, the P-I characteristic curve is linear. The outgoing light of the LED is input to the photodetector and its light intensity can be converted into a voltage signal, as shown in Figure 5. It can be seen from the figure that the intensity of the output of the modulated light source is a high frequency sinusoidal signal with a frequency of 5 kHz, which satisfies the modulation requirements required in this paper.

The sensing unit includes a polarizer, a magneto-optic crystal, a beam splitter, and an analyzer. The modulated light emitted by the LED generates a deflection angle after passing through the sensing unit placed in the center of the magnetic field. After passing through the photodetector, the light intensity is converted into a voltage signal. After passing through the filter circuit, amplification circuit, and phase-shifting circuit, the output voltage signal results into four-channel signals. These four-channel signals are collected by the data acquisition card USB-2405 to the computer and signal processing is performed in the Labview program. The current to be measured is generated by the current source, and the sensing unit is placed in the center of the powered solenoid. The current range to be measured was 0–4 A. The standard current transformer as the calculation error standard in this experimental platform selects the zero-magnetic DC current transformer and the measurement error of the transformer is 0.1% which is less than the measurement error of the system.

When the current was 4 A, the output of one of the two optical paths is shown in Figure 6. Figure 6 compares the denoising effect of the two de-noising algorithms, which are signal correlation detection and double correlation detection. It can be seen that the output voltage fluctuations using double correlation detection are smaller, which indicates that the double correlation detection algorithm can filter out noise effectively and improve the SNR.

When the current changes from 0 A to 4 A, the measurement error of the system is shown in Figure 7. As can be seen from Figure 7, the double correlation detection algorithm has a significant denoising effect that can reduce the measurement error and can also control the measurement error to about 0.3%, as compared with the single correlation detection algorithm.

Figure 8 compares the output voltages of the system when the TGG crystal and the ZF-7 magneto-optical glass are used as sensing materials. It can be seen that, the output voltages of the two materials have good linearity. When the sensing unit uses the TGG crystal as a sensing material, the slope of the output characteristic curve is significantly increased, and the amplitude of the output voltage is also significantly increased, which contributes to the improvement of the measurement sensitivity and the output SNR. The experimental measurement error is shown in Figure 9. As can be seen from the figure, by increasing the SNR of the output signal, the measurement error of the system can be effectively reduced.

## 6. Conclusions

In order to address the problems of a low SNR and the overlap of signal-to-noise band when DC current measurement is applied to OCTs, according to the Faraday magneto-optical effect and correlation detection principles, this paper proposed a DC measurement method based on TGG crystal and double correlation detection algorithm. The main conclusions are as follows:

(1) Using the TGG crystal as the sensing material can significantly increase the Faraday deflection angle and improve the SNR of the output signal. At the same time, the high Verdet constant of the TGG crystal increased the measurement sensitivity of the sensing unit.

(2) The theories of signal correlation detection algorithm and double correlation detection algorithm were deduced. Through simulation, the denoising effects of the two algorithms were compared. The analysis results showed that, as compared with the single correlation detection algorithm, the double correlation detection algorithm can further filter white noise, reduce fluctuations, improve the SNR, and reduce the measurement error.

(3) A hardware test platform was built for DC measurement. The signal processing program was written using the Labview software. Through theoretical and experimental analysis, it was verified that the DC measurement method based on the TGG crystal with high Verdet constant and the double correlation detection algorithm with better denoising effect can effectively reduce the measurement error and improve the measurement sensitivity.

## Figures and Tables

**Figure 1 sensors-19-02997-f001:**
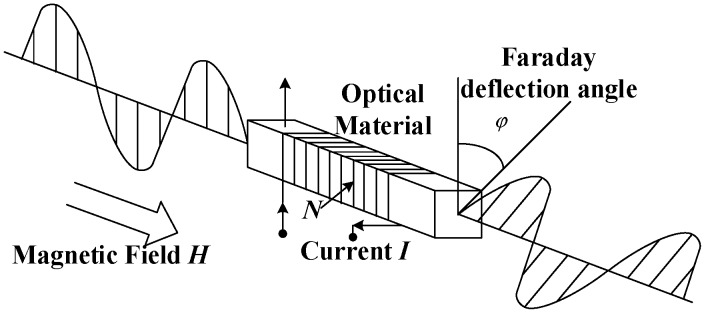
Faraday effect principle.

**Figure 2 sensors-19-02997-f002:**
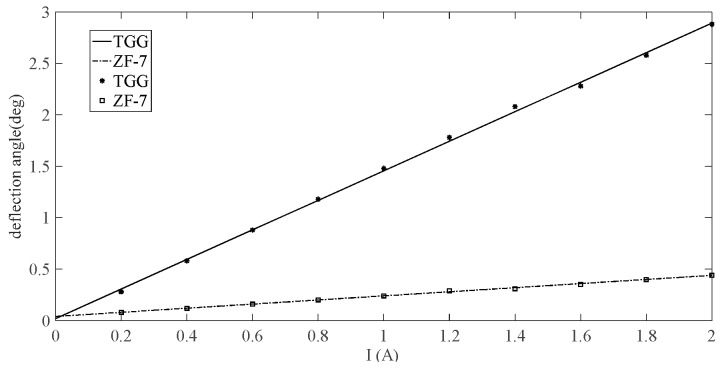
Faraday rotation experiment curves.

**Figure 3 sensors-19-02997-f003:**
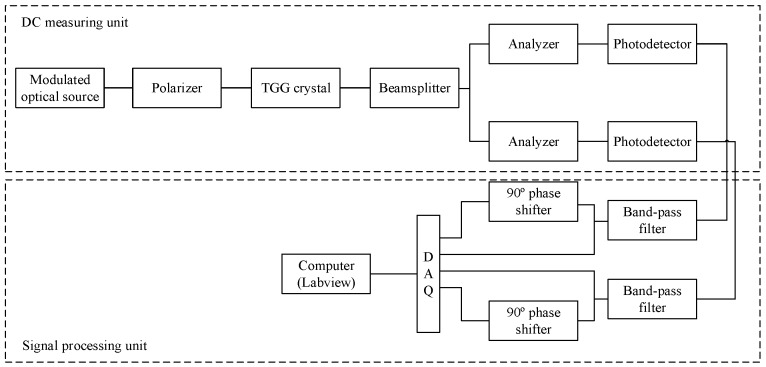
Block diagram of DC measurement system.

**Figure 4 sensors-19-02997-f004:**
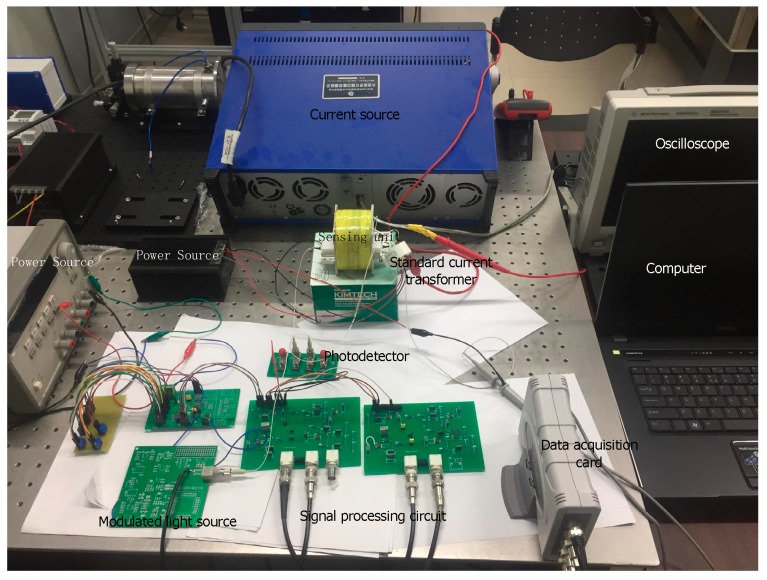
DC measurement experiment platform.

**Figure 5 sensors-19-02997-f005:**
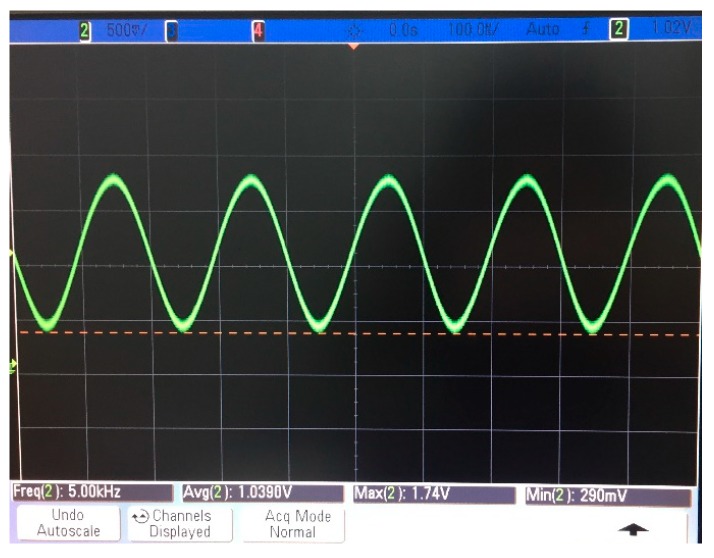
Waveform of the modulation light.

**Figure 6 sensors-19-02997-f006:**
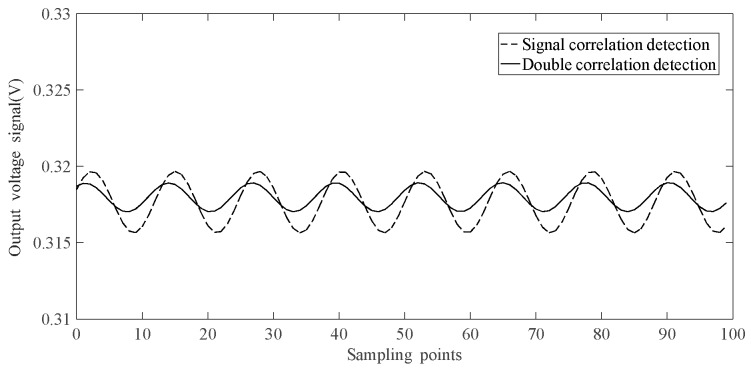
Output of one optical path.

**Figure 7 sensors-19-02997-f007:**
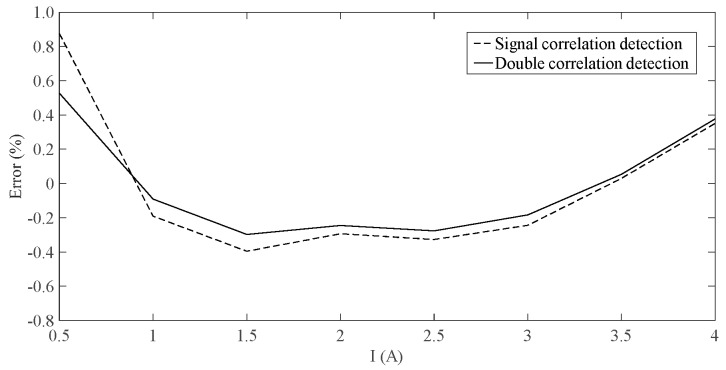
Error comparison.

**Figure 8 sensors-19-02997-f008:**
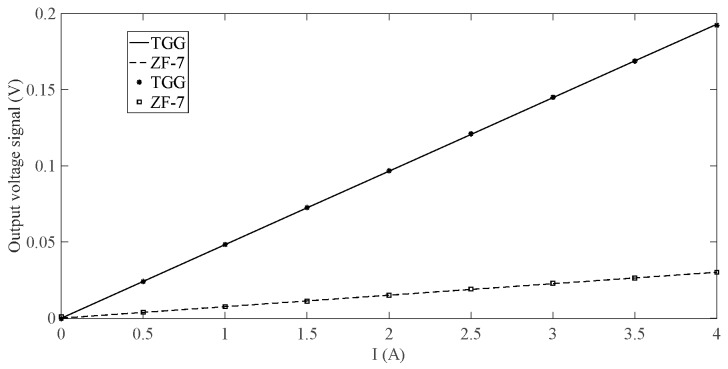
Measurement curves of TGG and ZF-7.

**Figure 9 sensors-19-02997-f009:**
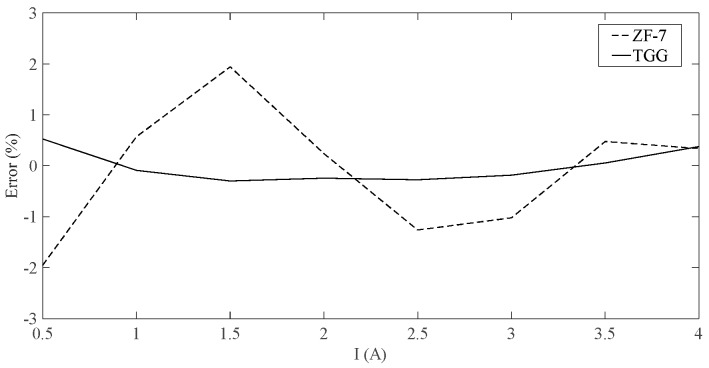
Comparison of measurement errors of TGG and ZF-7.

**Table 1 sensors-19-02997-t001:** Measurement errors of different signal processing methods.

	Signal Processing	Synchronous Demodulation	Single Correlation Detection Algorithm	Double Correlation Detection Algorithm
SNR				
0.2/1.0		0.050248	0.0121	0.00942
0.5/1.0		0.004286	0.00613	0.004242
1.0/1.0		0.00394	0.00333	0.002406

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
