# Peer review of "A Direct Current Measurement Method Based on Terbium Gallium Garnet Crystal and a Double Correlation Detection Algorithm"

_sensors, 2019, doi:10.3390/s19132997_

Round 1

Reviewer 1 Report

1) In Figure 1 it is necessary to show not only the direction of the magnetic field intensity H, but also the coil with N turns and current i.

2) It is necessary to explain in lines 115-116 that the dependencies shown in Figure 2 are obtained at the same number of turns N.

3) In line 163, it is necessary to make an explanation similar to what is done in lines 149-151.

4) The comma must be at the end of line 195 rather than at the beginning of line 196.

5) In the formula (25) is extracted from V’o1 and V’o2 amplitudes it is necessary to designate other letters.

Reviewer 2 Report

The manuscript is well crafted with clear information. The article profile is suitable for publication in Sensors.

Author Response

Thank you for reviewing my paper and your evaluation of my paper. Wish everything goes well with your work!

Reviewer 3 Report

Generally, the paper is short of novelty and the presentation is in need of further polishing. The author need to make it more professional.

1, Please define DC in the first paragraph. 

2, Please use OCT instead of the full name if it is already defined. Please pg through the paper for the semilar problems;

3, The curves in the picture is too thin, please revise and make it more visiable;

4, Please use the screenshot picture for Fig. 5 rather than such a captured picture. By the way, I cannot find any useful information in the picture, which can relate to the claim of "the modulated light is good" and the claim here should be more specific.
